# Impact of Spectral Resolution and Signal-to-Noise Ratio in Vis–NIR Spectrometry on Soil Organic Matter Estimation

**Bo Yu** [1,2], **Jing Yuan** [1,*], **Changxiang Yan** [1,3], **Jiawei Xu** [1,2], **Chaoran Ma** [1,2] and **Hu Dai** [4]

1. Changchun Institute of Optics, Fine Mechanics and Physics, Chinese Academy of Sciences, Changchun 130033, China; yubo19@mails.ucas.ac.cn (B.Y.); yancx@ciomp.ac.cn (C.Y.); xujiawei211@mails.ucas.ac.cn (J.X.); machaoran21@mails.ucas.ac.cn (C.M.)
2. University of Chinese Academy of Sciences, Beijing 100049, China
3. Center of Materials Science and Optoelectrics Engineering, University of Chinese Academy of Sciences, Beijing 100049, China
4. National Key Laboratory of Science and Technology on Vacuum Technology and Physics, Lanzhou Institute of Physics, Lanzhou 730000, China; daihulove1@126.com
*   Correspondence: yuanjing@ciomp.ac.cn; Tel.: +86-131-0440-2340

**Abstract:** Recently, considerable efforts have been devoted to the estimation of soil properties using optical payloads mounted on drones or satellites. Nevertheless, many studies focus on diverse pretreatments and modeling techniques, while there continues to be a conspicuous absence of research examining the impact of parameters related to optical remote sensing payloads on predictive performance. The main aim of this study is to evaluate how the spectral resolution and signal-to-noise ratio (SNR) of spectrometers affect the precision of predictions for soil organic matter (SOM) content. For this purpose, the initial soil spectral library was partitioned into to two simulated soil spectral libraries, each of which were individually adjusted with respect to the spectral resolutions and SNR levels. To verify the consistency and generality of our results, we employed four multiple regression models to develop multivariate calibration models. Subsequently, in order to determine the minimum spectral resolution and SNR level without significantly affecting the prediction accuracy, we conducted ANOVA tests on the RMSE and $R^2$ obtained from the independent validation dataset. Our results revealed that (i) the factors significantly affecting SOM prediction performance, in descending order of magnitude, were the SNR levels > spectral resolutions > estimation models, (ii) no substantial difference existed in predictive performance when the spectral resolution fell within 100 nm, and (iii) when the SNR levels exceeded 15%, altering them did not notably affect the SOM predictive performance. This study is expected to provide valuable insights for the design of future optical remote sensing payloads aimed at monitoring large-scale SOM dynamics.

**Keywords:** soil organic matter; optical remote sensing; spectrometers; multiple regression model; spectral resolution; signal-to-noise ratio; analysis of variance

## 1. Introduction

Soil forms the foundation of the agricultural ecological system. Soil organic matter (SOM) comprises a complex mixture of organic materials derived from plants and animals in different stages of decomposition. It exerts a profound influence on soil nutrients, plant development, human well-being, and the climate [1]. A decrease in SOM significantly influences soil structural stability, water retention, infiltration capacity, nutrient holding, soil biodiversity, fertility, and ultimately, ecological and agroecological productivity. Precisely assessing the SOM holds immense importance for food production, carbon cycling, and climate regulation [2]. Conventional methods for estimating the SOM traditionally rely on labor-intensive field soil sampling and subsequent laboratory analysis. These approaches are expensive, time-consuming, destructive, and have limited spatial coverage. In contrast, spectroscopy technology offers distinct advantages, including high efficiency,

speed, non-destructive detection, and ease of use. Over the past few years, with the rapid development of spectral sensing technology, numerous researchers have been utilizing visible and near-infrared (Vis–NIR) spectroscopy technology to estimate SOM content [3,4]. Furthermore, spectral sensors can be mounted on unmanned aerial vehicles and satellites to achieve the purpose of real-time monitoring of the dynamic change in the SOM in a large area [5–8]. Consequently, the utilization of spectroscopic techniques for SOM content determination holds significant importance in achieving precision agriculture and in advancing agricultural modernization.

The spectral profile of soil results from the combination of various soil constituents, which is characterized by its nonspecific, faint, and broad nature due to the overlap of absorption bands and the low concentrations of soil components. Therefore, multivariate calibration techniques are usually used to build the relation between the SOM content and soil diffuse reflectance spectra [9]. Common multivariate linear regression models include multiple linear regression, ridge regression (RR), principal component regression (PCR), partial least squares regression (PLSR), and so on [8,10–12]. Linear regression models provide more interpretability. To be specific, RR, PCR, and PLSR are insensitive to collinearity due to their mathematical principles. Moreover, the relationship between the dependent variable (SOM content) and the spectral data may not be purely linear due to the complex composition of soil properties. Machine learning methods, such as support vector machine regression, (SVMR), the back propagation neural network (BPNN), the cubist regression tree, random forests, and others, can be employed to address nonlinear problems [8,13,14]. The inherent structures of these methods makes it challenging for them to uncover the functional relationship between Vis–NIR spectra and the SOM. Numerous studies have utilized various multiple regression models to predict the SOM. However, there is no consensus on which multiple regression model yields optimal prediction performance due to variations in soil categories and measurement environments. Therefore, to validate the consistency and generality of our findings, four regression techniques (RR, PLSR, SVMR, and BPNN) were used in this study to demonstrate the effects of variations in two core parameters (spectral resolution and signal-to-noise ratio (SNR) levels) of the spectrometer on the estimation of SOM.

Different spectral treatments and modeling approaches can affect prediction accuracy [15,16]. Nevertheless, prediction accuracy is also contingent on the parameters of Vis–NIR spectrometers. In general, a sensor with more spectral bands, higher spectral resolution, and a SNR tends to produce more accurate data, albeit with potential data redundancy. It is a well-known fact that hyperspectral data exhibit autocorrelation, meaning that many wavelengths convey the same information about land cover properties. Can an abundance of wavelengths, high spectral resolutions, and elevated SNR levels significantly enhance SOM prediction accuracy? Or, how should we select spectral resolution and the SNR to efficiently estimate the SOM? Currently, there is limited research analyzing the influence of spectrometer core parameters on soil property estimation. Castaldi et al. reported that introducing noise into the simulated spectra resulted in a decrease in the prediction accuracy of the model. They also found that a spectral resolution of 40 nm could yield soil texture estimation accuracy that was quite comparable to what sensors with higher spectral resolutions achieved [17]. Knadel et al. conducted a comparative analysis of the prediction performance using three Vis–NIR spectrometers that differed in spectral resolutions, SNRs, and spectral bands. Their findings indicated that the spectral range had the most significant impact on the prediction performance. Moreover, they emphasized that when considering the trade-off between the spectral resolution and SNR, a high SNR played a more crucial role [18]. Gomez et al. established ten spectral configurations featuring various spectral resolutions. Their findings revealed that the spectral configurations within the spectral resolution range of 5 to 100 nm delivered comparable and effective predictive performance for clay estimation [19]. Jia transformed airborne hyperspectral images into degraded hyperspectral libraries with different spectral resolutions, spatial resolutions, and SNR levels. They subsequently assessed the classification accuracy of these degraded

hyperspectral datasets for crop identification. The findings indicated that the accuracy declined as the SNR levels decreased. Regarding spectral resolution, the accuracy exhibited an initial increase, followed by stabilization and ultimately a decline [20]. This study builds upon prior research and determines the ideal instrument parameters for accurate SOM estimation.

The purpose of this work is to investigate how the spectral resolution and SNR impact the accuracy of SOM estimation. We created two spectral libraries with varying spectral resolutions and SNR levels using the initial spectral library. This allowed us to individually manipulate the spectral resolution and SNR level. To ensure the consistency and generalizability of our findings, we employed four regression models (RR, PLSR, SVMR, and BPNN) with spectral data featuring different spectral resolutions and SNR levels as input variables. We utilized the analysis of variance (ANOVA) technique to further explore which factors significantly influenced the SOM predictive performance, and we identified the lowest spectral resolution and SNR level that did not significantly affect the prediction accuracy.

## 2. Materials and Methods

### 2.1. Study Area and Soil Data Collection

The study was carried out in Qiqihar, Heilongjiang Province, Northeast China, which extends from 109°26′15.86″ to 126°45′22.44″ E and from 47°24′31.40″ to 48°14′25.33″ N (Figure 1). The study area has a typical temperate continental monsoon climate with four distinct seasons. Generally, the temperature varies between −24 °C and 27 °C and rarely goes below −29 °C or above 32 °C during a year. The annual average rainfall varies from 400 to 550 mm. The soil type in the study area is mainly black soil, which has the characteristics of high heat, good permeability, light texture, and high soil organic carbon content [21]. Corn, soybeans, and rice are presently the predominant crops grown in this region

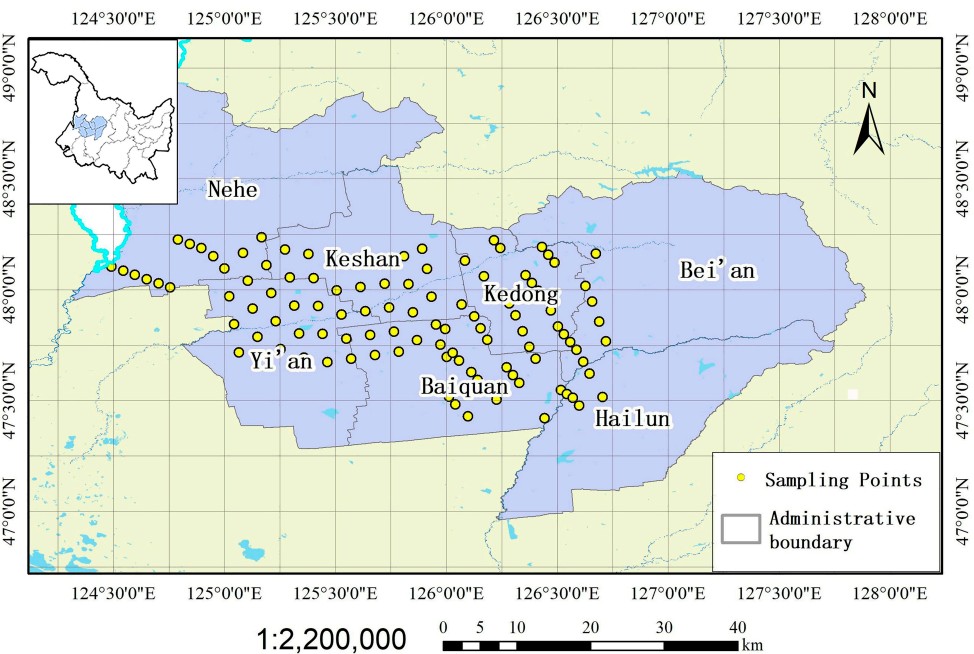

**Figure 1.** Location of the study in China.

In October 2016, we randomly gathered 112 soil samples from the surface layers (0–15 cm). In the field, we removed the surface litter, collected the soil samples, and sealed them in plastic bags. Each sampling location was recorded using a portable GPS. Back in the laboratory, we allowed the soil samples to air-dry naturally, gently ground them, and sifted them through a 1 mm sieve to obtain the fine earth fraction, eliminating small stones,

coarse roots, and fallen leaves. Subsequently, we conducted spectral analysis and chemical determinations on the processed soil samples in the laboratory.

## 2.2. Analysis of SOM Content and Spectra Measurement

A conversion coefficient of 1.724 was applied to convert the soil organic carbon (SOC) to SOM using the formula SOM (g kg$^{-1}$) = 1.724 × SOC (g kg$^{-1}$). In our study, SOC was measured by the potassium dichromate oxidation–outer heating method. Vis–NIR diffuse reflectance spectra of soil samples were measured in a dark room using an ASD FieldSpec3 (Analytical Spectral Devices, Boulder, CO, USA) with a spectrum range of 350 to 2500 nm. For the spectral range of 350 to 1000 nm, the spectral sampling interval of the ASD spectroradiometer is 1.4 nm, thus providing a spectral resolution of 3 nm. In the range of 1000 to 2500 nm, the spectral sampling interval is 2 nm, with a spectral resolution of 10 nm. The reflectance data available to users were resampled with ViewSpecPro (version 6.0.0, ASD, Boulder, CO, USA) across both spectral ranges, thus resulting in 2151 spectral bands. We used a 50 W halogen lamp with a 30° incident angle as a light source, located at a distance of 10 cm from the soil samples. An optical probe with a field of view (FOV) of 1° was vertically placed 5 cm from the center of the soil sample's surface. Reflectance was calibrated using a normalized white panel before the readings commenced and every 30 min thereafter. To reduce noise, we conducted ten measurements for each soil sample and then averaged them to obtain the spectra. Splicing correction in the ViewSpecPro software was applied to resolve the breakpoints near 1000 nm and 1800 nm. The spectral regions of 350–450 nm and 2401–2500 nm were omitted due to significant instrumental artifacts at the edges of the spectrum, thus resulting in a total of 1951 spectral bands.

## 2.3. Creation of Simulated Spectral Libraries

To investigate the effect of two core parameters, namely, spectral resolution and SNR, of spectrometers on the SOM prediction performance, we transformed the initial soil spectral library (comprising soil spectra measured by an ASD spectrometer) to create two simulated spectral libraries. The first library, referred to as the spectral configuration library, consists of nine degraded spectral configurations with regular spectral resolution (i.e., the spectral resolution remains constant throughout the considered spectral field). The number of spectral bands in these spectral configurations was reduced from 323 to 8, and the spectral resolution was coarsened from 3 and 10 nm for the 450–1000 nm and 1000–2400 nm, respectively, to 200 nm (Table 1). The second library, known as the spectral SNR library, involved reducing the SNR of the spectral data to the desired SNR level by introducing Gaussian noise to the original spectra. This SNR library contains a total of 15 spectral dataset, spanning SNR values from 100% down to 1%. By separately simulating these two spectral libraries, we can assess the sensitivity of SOM prediction results to spectral resolution and SNR, independent of other variables. The process of generating these two simulated spectral libraries is detailed in the following section.

**Table 1.** Descriptions of ten spectral configurations. ASD_1/1 represents the initial spectral library.

| Configurations | N | 450–1000 nm | | 1000–2400 nm | |
|---|---|---|---|---|---|
| | | SSI | SR | SSI | SR |
| ASD_1/1 | 1951 | 1 | 3 | 1 | 10 |
| Con_3/10 | 323 | 3 | 3 | 10 | 10 |
| Con_10/10 | 194 | 10 | 10 | 10 | 10 |
| Con_20/20 | 96 | 20 | 20 | 20 | 20 |
| Con_40/40 | 47 | 40 | 40 | 40 | 40 |
| Con_60/60 | 31 | 60 | 60 | 60 | 60 |
| Con_80/80 | 23 | 80 | 80 | 80 | 80 |
| Con_100/100 | 18 | 100 | 100 | 100 | 100 |
| Con_150/150 | 12 | 150 | 150 | 150 | 150 |
| Con_200/200 | 8 | 200 | 200 | 200 | 200 |

### 2.3.1. Spectral Configuration Library

The spectral configuration library contains a total of 10 spectral configurations, which were divided into two categories: ASD_1/1 and 9 Con_X/Y (Table 1). The original soil spectra were named ASD_1/1 because the original spectra measured by the ASD spectrometer were sampled at intervals of 1 nm in the range of 450–1000 nm and 1000–2400 nm. The other 9 Con_X/Y configurations were derived from ASD_1/1, where X represents the spectral resolution from 450 to 1000 nm, and Y represents the spectral resolution from 1000 to 2400 nm. Con_X/Y reduces the number of spectral bands from 323 to 8, decreases the spectral resolution from 3 nm to 200 nm, and sets the spectral sampling interval equal to the spectral resolution. The spectral reflectance of Con_X/Y can be calculated as follows:

Firstly, three parameters are defined in the spectral configuration [22]: the number of spectral bands (N), the spectral resolution (SR), and the spectral sampling interval (SSI). The SR is also called the full half-width maximum (FHWM), and SSI represents the interval between the acquisition of two signals (Figure 2). Then, the initial laboratory-measured spectra are resampled with Gaussian filters whose tails are cut to twice their width, following the filter response function G ($\lambda$):

$$G(\lambda) = \exp\left(\frac{-(\lambda - \lambda_c)^2}{2 \cdot \sigma^2}\right) \text{with} \sigma = \frac{SR}{2 \cdot \sqrt{2 \cdot \ln(2)}} \tag{1}$$

where $\lambda$ represents the spectral step under different SSIs; $\lambda_c$ denotes the central waveband within the range of the spectral response to certain SSIs.

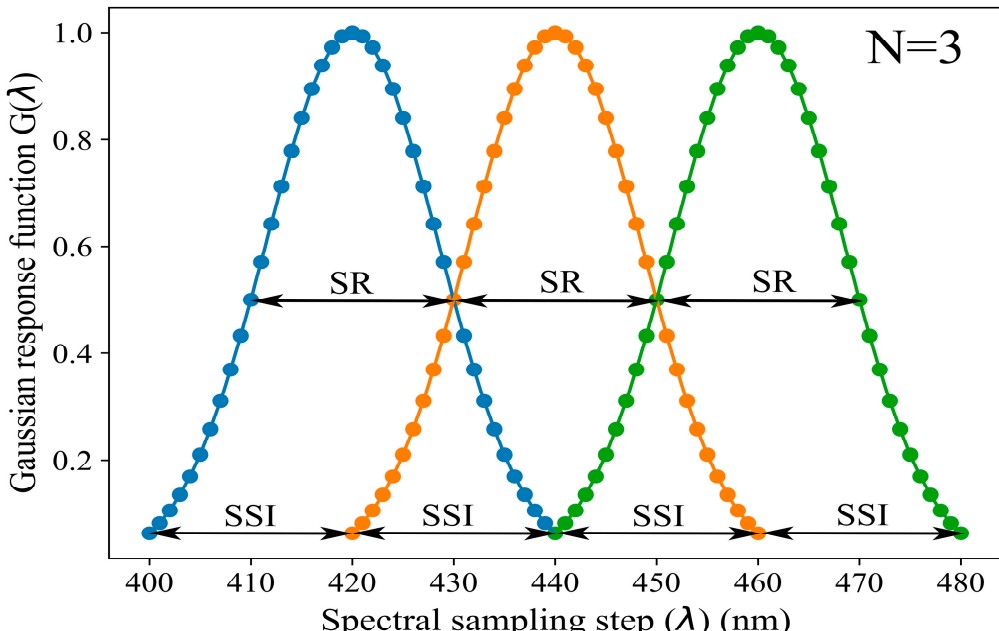

**Figure 2.** The Gaussian response function G($\lambda$) for N = 3, SR = 20 nm, and SSI = 20 nm.

Finally, the spectral reflectance of each Con_X/Y is determined as follows:

$$R_i = \frac{\sum_{\lambda_s}^{\lambda_k} r(\lambda) \cdot G(\lambda)}{\sum_{\lambda_s}^{\lambda_k} G(\lambda)} \tag{2}$$

where $\lambda_s$ and $\lambda_k$ represent the spectral reflectance of the starting and ending bands within the spectral range, respectively.

### 2.3.2. Spectral SNR Library

The SNR is an important factor affecting the performance of a spectrometer, which compares a signal value in the presence of a signal with a value for system noise in the absence of a signal. In fact, due to the different principles and operating environments of various spectrometers, the models for calculating the SNR differ. To study the impact of SNR on SOM estimation performance, it is necessary to choose a reasonable range of SNR values. In this study, the SNR of each soil spectrum was calculated as the ratio of the mean value of the reference signal intensity to the standard deviation [23] (Figure 3). In this study, the mean and standard deviation of spectral reflectance were obtained from 10 repeated measurements of a soil sample.

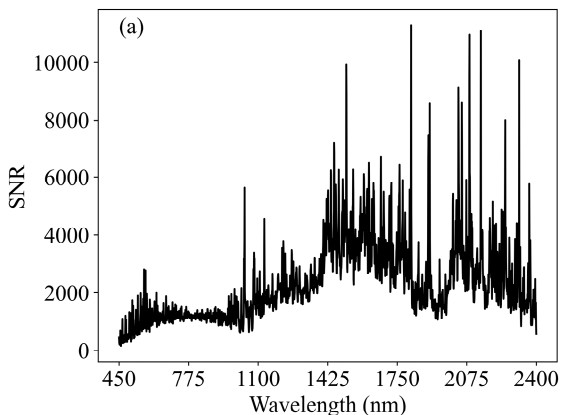
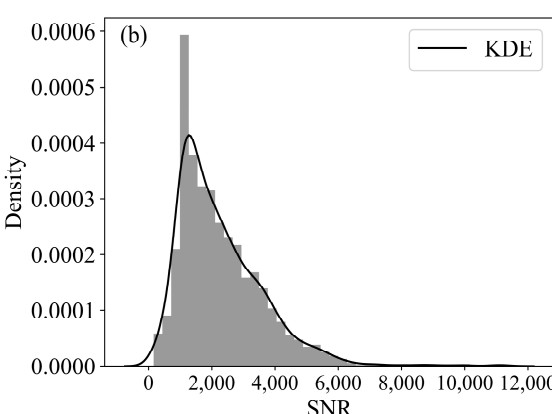

**Figure 3.** (**a**) The SNR of a soil sample spectrum with a total of 1951 spectral bands in the spectral range from 450 nm to 2400 nm. (**b**) Histogram and probability density estimation of SNR using KDE.

We repeated this operation for all spectral bands of the soil spectrum and obtained SNRs for a total of 1951 bands from 450 nm to 2400 nm (Figure 3a). To better understand the distribution of SNR data, we plotted a histogram of SNR data and estimated the probability density function using the kernel density estimation (KDE) method (Figure 3b). KDE can be seen as the smoothing result of the histogram and allows a better representation of multimodality [24]. In this study, the Gaussian kernel was chosen as the kernel of KDE. We set up a spectral SNR library containing 15 different SNRs from 100% to 1% SNR levels (100%, 90%, 80%, 70%, 60%, 50%, 40%, 30%, 25%, 20%, 15%, 10%, 5%, 3%, and 1%). To simulate the spectra with different SNR levels, we added random Gaussian noise to each initial spectral. The process can be represented by the following equations [25]:

$$N\,(L) = \frac{L}{SNR\,(L)} \tag{3}$$

$$N_d\,(L) = \sqrt{N_i\,(L)^2 - N\,(L)^2} \tag{4}$$

$$L_d = L + Rnd\,(0,1) \cdot N_d\,(L) \tag{5}$$

where $L$ represents the measured raw radiance, $N_d\,(L)$ represents the noise added to the soil spectrum to degrade the SNR to the required noise $N_i\,(L)$, and $Rnd(0,1)$ is used to generate a Gaussian distribution that has a mean value of 0 and a standard deviation of 1. $L_d$ is the radiance after adding Gaussian noise.

In practice, the SNR $(L)$ is calculated for each band as shown in Figure 3a, and then the original noise level $N\,(L)$ for radiance $L$ is obtained using Equation (3). Similarly, the noise $N_i\,(L)$ corresponding to the different desired degradation SNRs for radiance $L$ can also be calculated using Equation (3). The noise $N_d\,(L)$ to be added to the spectral data can then be calculated using the fact that noise can add in an orthogonal manner (i.e., the square of the noise can be considered as the sum of each individual noise source) as shown

in Equation (4). Finally, the radiance $L_d$ with different SNRs can be obtained in terms of the original radiance L and the added noise Rnd $(0, 1) \cdot N_d$ (L) according to Equation (5).

### 2.4. Calibration and Validation of Prediction Models

The whole soil data set was randomly split into a calibration set and a validation set at the ratio of 7:3. To more accurately assess the effect of instrumental parameters on SOM estimation, the above process of partitioning the soil data set was repeated 20 times. Levene's test was performed on both the calibration sets and validation sets for each group to ensure that they had the same data distribution. Four multivariate techniques (RR, PLSR, SVMR, and BPNN) were used to build the relation between the spectra and SOM on the calibration sets and to test them on the independent validation sets. Before models' development, the raw spectra were subjected to the Savitzky–Golay smoothing (SG) smoothing with a second-order polynomial and window size of 11 wavelengths, including first derivative (FD), second derivative (SD), absorbance (Abs), and multiplicative scatter correction (MSC) and their combinatorial operations. Finally, normalization (NOR) was carried. The prediction accuracy of the data using the normalized spectra achieved the desired prediction results. In addition, our study mainly focused on the impact of spectral resolution and SNR, two crucial parameters of spectroscopic instruments, on the accuracy of SOM estimation, rather than on enhancing the prediction accuracy of the models. Therefore, the we only normalized the raw spectra before modeling. A brief description of each multiple regression technique is provided below.

Most of the spectral variables in the full Vis–NIR spectrum exhibit strong correlations, indicating that the reflectance at one wavelength is very similar to the value at the adjacent wavelength. Although RR and PLSR are linear models, they are particularly well-suited for addressing multicollinearity issues due to their mathematical principles and are therefore commonly used for the analysis of Vis–NIR spectroscopy data. RR is a modified least squares estimation approach that applies a penalty on the size of the coefficients [26]. We controlled the hyperparameter $\alpha$ to regulate the importance of features. A larger $\alpha$ value indicates a stronger regularization. Typically, the optimal $\alpha$ strikes a balance between calibration model prediction performance, introducing slight bias to the regression while effectively addressing multicollinearity. PLSR integrates compression and regression steps by extracting a fixed number of orthogonal factors called latent variables (LVs) to maximize the covariance between the predictor and response variables [27]. Therefore, the essence of PLSR lies in creating an appropriate linear combination of features that carry the most information rather than processing a large set of correlated data. SVMR is a kernel-based modeling technique rooted in statistical learning theory. It projects raw data from a low-dimensional feature space into a high-dimensional feature space using an implicit mapping (also called a kernel function). It constructs an optimal linear hyperplane as a decision function for nonlinear regression problems and then inversely transforms in the nonlinear space. In summary, SVMR simplifies the problem by converting nonlinear regression into a linear problem in a high-dimensional feature space [28]. BPNN is a highly nonlinear mathematical model composed of nodes (or neurons) organized in layers and connected by links. BPNN optimizes the weights between neurons based on the backpropagation of errors, minimizing backward learning error from the output layer to the input layer [29].

For these four types of multiple regression models mentioned above, it is necessary to select appropriate hyperparameters to enhance the models' performance. The $\alpha$ value for RR is set in a geometric progression ranging from 0.001 as the minimum to 1 as the maximum, with a total of 30 values. The maximum number of LVs for PLSR was set to 15. In the case of SVMR, we opted for the Gaussian radial basis function (RBF) as the kernel function, which involves two critical hyperparameters: regularization parameters C and $\gamma$. We set the values for C as 0.1, 1, 10, 50, 80, 100, 120, 150, 180, and 200 and $\gamma$ as 0.0001, 0.001, 0.01, 0.1, and 10. For BPNN, we constructed a 4-layer perceptron network with one input layer (spectra data), two hidden layers (the number of nodes per layer was set to 2, 4, 6, . . ., 20), and an output layer. We chose the Relu function as the activation function for

the hidden layer, and L- BFGS was selected as the solver for weight optimization. In this study, a grid search and 5-fold cross-validation were employed on the calibration sets to determine the optimal hyperparameters for each model that minimized the cross validated root mean square error ($RMSE_{CV}$). All four multivariate regression models in this study were implemented in Python 3.8.5 and the scikit-learn package.

### 2.5. Evaluation of Models

The root mean squared error (RMSE), the coefficient of determination ($R^2$), and the ratio of performance to interquartile distance (RPIQ) are employed to assess the prediction performance of the four models in both calibration sets and validation sets.

$$RMSE = \sqrt{\frac{\sum_{i=1}^{N}(y_i - \hat{y}_i)^2}{N}} \tag{6}$$

$$R^2 = 1 - \frac{\sum_{i=1}^{N}(y_i - \hat{y}_i)^2}{\sum_{i=1}^{N}\left(y_i - \bar{y}\right)^2} \tag{7}$$

where $y_i$ and $\hat{y}_i$ represent the observed values and the predicted values, respectively, $\bar{y}$ is the mean of the observed values, and N is the number of samples with i ranging from 1 to N.

$$RPIQ = \frac{IQ}{RMSE} \tag{8}$$

where IQ represents the difference between the third and first quartiles.

RMSE quantifies the difference between observed values and predicted values, measured in the same units as the dependent variable. $R^2$ signifies the proportion of variance in the dependent variables explained by the independent variables in the regression models, and it is utilized to assess the goodness of fit of the models. RPIQ accounts for both prediction error and the variation in observed values, providing a more objective and easily comparable measure of model validity during model validation [30,31]. In our study, $RMSE_{cv}$, $R^2_{cv}$, and $RPIQ_{cv}$ represent the RMSE, $R^2$ and RPIQ in cross-validation, respectively. $RMSE_p$, $R^2_p$, and $RPIQ_p$ represent the RMSE, $R^2$ and RPIQ in independent validation sets, respectively.

### 2.6. Application of ANOVA Technique

ANOVA is a statistical method used to compare whether the means of two or more groups differ significantly and to assess the variance using a probability distribution [32]. It enables us to dissect the variation attributable to each factor in relation to the total variation in the presence of random interference. ANOVA determines whether the variation significantly affects the study population and rejects the null hypothesis. This is done by comparing F values with the critical F value ($F_{crit}$) to recognize the significance of each factor's contribution. Additionally, the *p* value can be computed to determine whether a factor exerts a significant influence on the research subject. The *p* value indicates the probability of the null hypothesis $H_0$ being valid. $H_0$ posits that that there is no difference among the studied groups, while $H_1$ suggests that a difference exists. We use the *p* value to decide whether reject $H_0$ by comparing it to the significance level, which represents the probability of rejecting the null hypothesis when it is true. Therefore, a smaller *p* value implies a higher likelihood of H1 being correct [33].

In this study, we employed ANOVA to assess the impact of the spectral resolution and SNR on the accuracy of SOM estimation. Since different prediction models may yield varying SOM estimates, we conducted two-way ANOVAs for spectral configuration prediction models and SNR level prediction models. To delve deeper into the influence of instrument parameters on SOM prediction performance, this study proceeded to conduct one-way ANOVA under each of the four prediction models to determine whether different

spectral configurations had a significant effect on the accuracy of SOM estimation. The same approach was used to analyze the effect of SNR on SOM prediction performance.

## 3. Results

### 3.1. Analysis of SOM

The SOM content in the study area ranged from 29.308 g kg$^{-1}$ to 59.650 g kg$^{-1}$, with a mean value of 42.780 g kg$^{-1}$, a standard deviation of 8.198, and a coefficient of variation of 19.16%, thus indicating a moderate level of variation. The entire soil data set ($n = 112$) was randomly divided into two portions: calibration sets, comprising 7/10 of the total soil samples ($n = 78$), and validation sets, comprising 3/10 of the total soil samples ($n = 34$), which was done to provide a more accurately assessment of the impact of the instrumental parameters on the SOM estimation. The aforementioned process of partitioning the soil dataset was repeated 20 times. The statistical description of the results of the 20 divisions of the soil data in this study is presented in Figure 4, and the outcomes of the Levine's variance test ($p > 0.8$) for all groups at a significance level of 0.05 indicated consistent variance between the calibration and validation sets (Table 2). Therefore, we assumed that the validation sets represented the data under investigation, and the calibrated model was utilized to predict the SOM content of the validation sets.

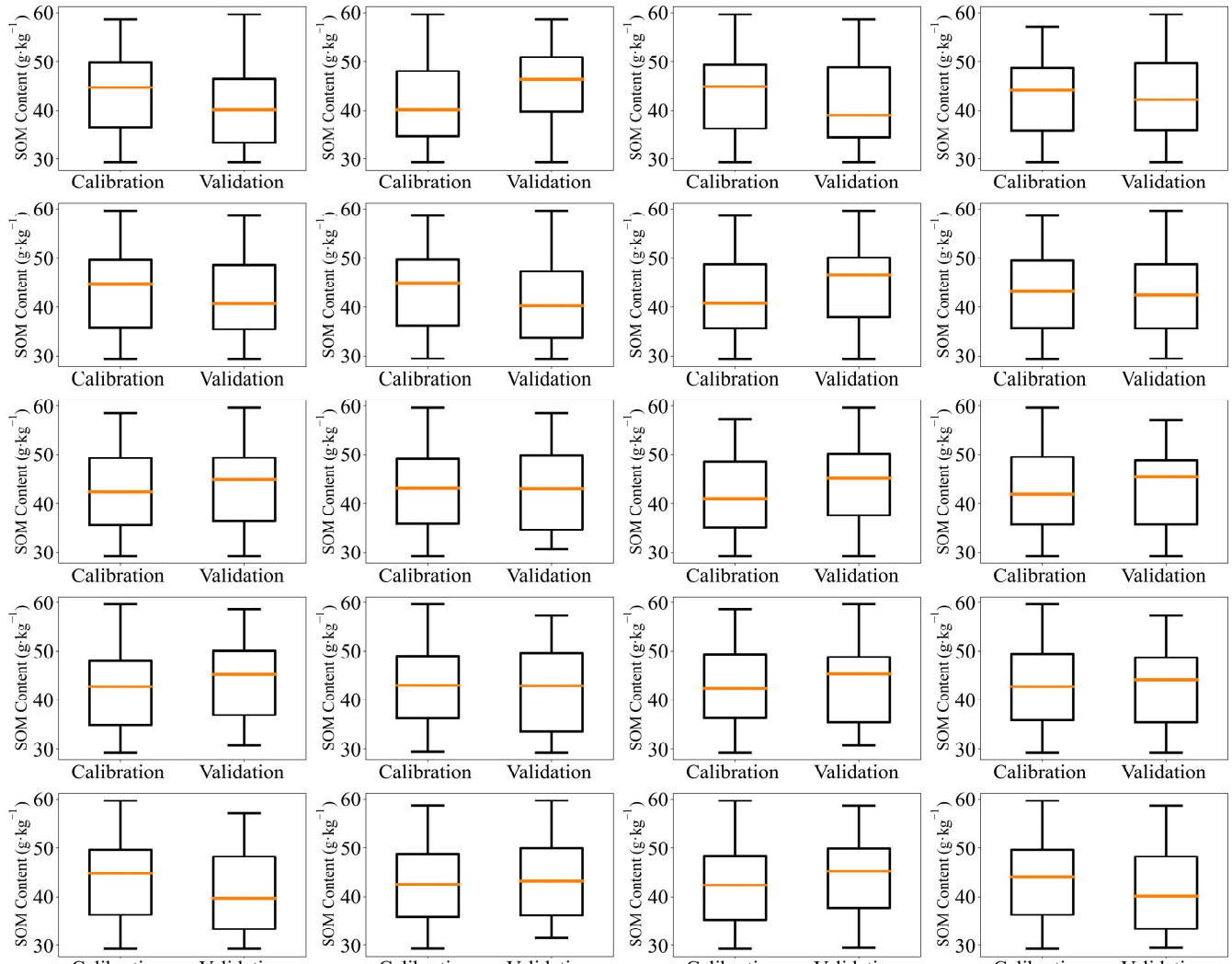

**Figure 4.** Boxplots depicting the SOM values for the calibration and validation sets across 20 groups.

**Table 2.** Descriptive statistics of SOM for the calibration and validation sets, along with the probability levels from Levene's test under various groupings.

| | Calibration Sets (g kg$^{-1}$) | | | | Validation Sets (g kg$^{-1}$) | | | | Levene's Test |
|---|---|---|---|---|---|---|---|---|---|
| | **Min** | **Max** | **Mean** | **SD** | **Min** | **Max** | **Mean** | **SD** | |
| 1 | 29.308 | 58.616 | 43.647 | 8.124 | 29.308 | 59.650 | 40.773 | 8.141 | 0.840 |
| 2 | 29.308 | 59.650 | 41.646 | 7.914 | 29.308 | 58.616 | 45.400 | 8.367 | 0.921 |
| 3 | 29.308 | 59.650 | 43.531 | 8.004 | 29.308 | 58.616 | 41.042 | 8.505 | 0.824 |
| 4 | 29.308 | 57.237 | 42.604 | 8.058 | 29.308 | 59.650 | 43.186 | 8.632 | 0.961 |
| 5 | 29.308 | 59.650 | 43.205 | 8.213 | 29.308 | 58.616 | 41.796 | 8.209 | 0.989 |
| 6 | 29.480 | 58.616 | 43.409 | 7.976 | 29.308 | 59.650 | 41.322 | 8.644 | 0.827 |
| 7 | 29.308 | 58.616 | 42.012 | 7.963 | 29.308 | 59.650 | 44.555 | 8.582 | 0.985 |
| 8 | 29.308 | 58.616 | 42.848 | 8.126 | 29.480 | 59.650 | 42.621 | 8.493 | 0.815 |
| 9 | 29.308 | 58.616 | 42.557 | 8.101 | 29.308 | 59.650 | 43.294 | 8.527 | 0.873 |
| 10 | 29.308 | 59.650 | 42.741 | 8.207 | 30.687 | 58.616 | 42.868 | 8.308 | 0.800 |
| 11 | 29.308 | 57.237 | 41.975 | 7.945 | 29.308 | 59.650 | 44.641 | 8.595 | 0.925 |
| 12 | 29.308 | 59.650 | 42.494 | 8.172 | 29.308 | 57.064 | 43.439 | 8.352 | 0.845 |
| 13 | 29.308 | 59.650 | 42.222 | 8.257 | 30.687 | 58.616 | 44.069 | 8.039 | 0.894 |
| 14 | 29.480 | 59.650 | 43.128 | 8.207 | 29.308 | 57.237 | 41.974 | 8.251 | 0.823 |
| 15 | 29.308 | 58.616 | 42.664 | 8.165 | 30.687 | 59.650 | 43.046 | 8.400 | 0.833 |
| 16 | 29.308 | 59.650 | 42.942 | 8.251 | 29.308 | 57.237 | 42.405 | 8.192 | 0.954 |
| 17 | 29.308 | 59.650 | 43.401 | 8.099 | 29.308 | 57.237 | 41.344 | 8.376 | 0.901 |
| 18 | 29.308 | 58.616 | 42.471 | 8.121 | 31.545 | 59.650 | 43.493 | 8.460 | 0.915 |
| 19 | 29.308 | 59.650 | 42.278 | 8.172 | 29.480 | 58.616 | 43.940 | 8.271 | 0.894 |
| 20 | 29.308 | 59.650 | 43.415 | 8.246 | 29.480 | 58.616 | 41.311 | 8.020 | 0.938 |

### 3.2. Spectral Characteristics of the Studied Soils

The spectral profiles of 112 soil samples (Figure 5) exhibited a decrease in reflectance as the SOM content increased, thereby aligning with findings from previous studies [34–36]. These soil spectra profiles exhibited a general similarity, which was characterized by lower reflectance in the visible (Vis) ranges and higher reflectance in the near-infrared (NIR) ranges. Notably, three distinctive absorption peaks emerged near 1400 nm, 1900 nm, and 2200 nm. The absorption peaks around 1400 nm and 1900 nm are associated with crystallized water and hydrated water [37]. The wavelengths near 2200 nm may be associated with organic molecules, Si-OH bonds, and cation-OH bonds within phyllosilicate minerals [38].

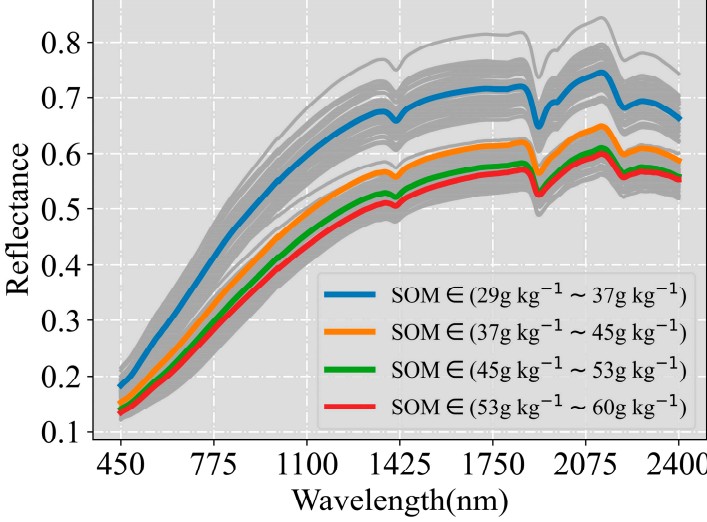

**Figure 5.** Reflectance spectra for the entire data set. The highlighted spectra curves correspond to the mean values of the spectra across different ranges of SOM contents.

### 3.3. Reflectance Spectrum of Spectral Configuration Library

The spectral reflectance profiles of the nine spectral configurations provide a more visually discernible representation of their distinctions (Figure 6). Upon visual inspection of these spectra, it becomes evident that as the spectral resolution decreased, differences in the profiles of the nine spectral configurations started to emerge, thereby resulting in variations in the magnitude of the spectral band reflectance.

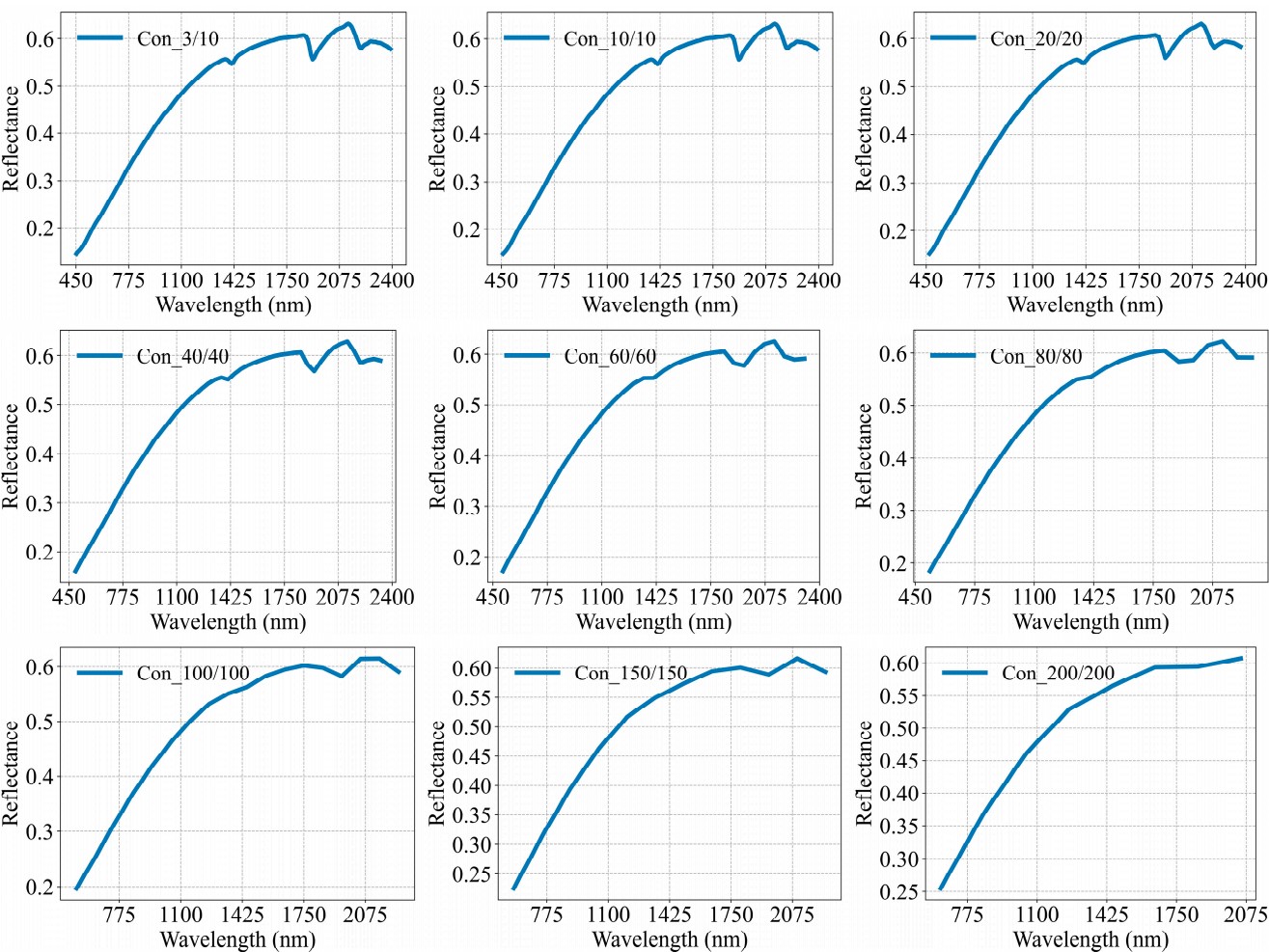

**Figure 6.** Reflectance spectra for each spectral configuration from Con_3/10 to Con_200/200.

### 3.4. Reflectance Spectrum of SNR Library

To investigate the effect of the spectrometer's SNR on the accuracy of the SOM estimation, we generated a spectral library with 15 different SNRs ranging from 1% to 100%, based on Equations (3)–(5). From Figure 7, we can conclude that when the SNRs were low, the spectral information of the objects became overwhelmed by noise, thus severely impacting the quality of the spectral data. As the SNRs increased, the spectra profiles gradually became clearer, and at high SNR levels, the spectral profiles appeared quite similar.

### 3.5. Prediction Results of Multiple Regression Models

We designed two independent experiments to analyze the effects of the spectral resolution and SNR on the accuracy of the SOM estimation. The evaluation results were derived from averaging the $RMSE_p$, $R^2_p$, and $RPIQ_p$ values across 20 independent tests. This section discusses the average statistics of these 20 groups. The $RMSE_{cv\_av}$, $R^2_{cv\_av}$, and $RPIQ_{cv\_av}$ represent the averages of the $RMSE_{cv}$, $R^2_{cv}$, and $RPIQ_{cv}$, respectively, across

the 20 groups. Similarly, the $RMSE_{p\_av}$, $R^2_{p\_av}$, and $RPIQ_{p\_av}$ represent the average of the $RMSE_p$, $R^2_p$, and $RPIQ_p$, respectively, across the 20 groups.

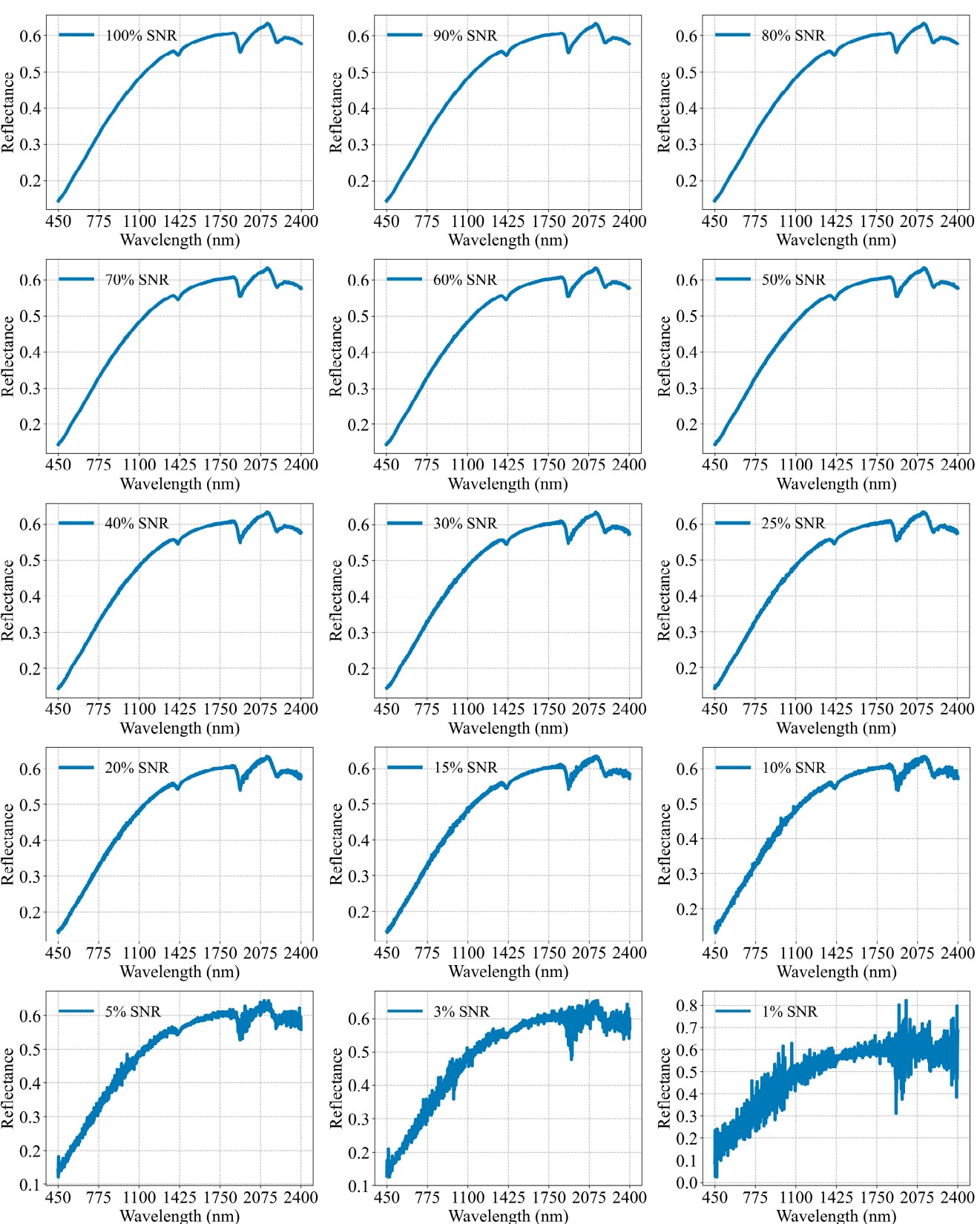

**Figure 7.** Reflectance spectra for different SNR levels from 1% SNR to 100% SNR.

In the first experiment, we focused on investigating how the coarsened spectral resolutions affected the SOM estimation. We built four prediction models for each spectral configuration, which ranged from ASD_1/1 to Con_200/200 nm in terms of the spectral resolution. The performance indicators (RMSE, $R^2$, and RPIQ) obtained with these four prediction models consistently exhibited the same trend concerning the spectral resolution variation (refer to Table 3). As we transitioned from ASD_1/1 to Con_100/100 nm, there was only a slight change in the $RMSE_{p\_av}$, $R^2_{p\_av}$, and $RPIQ_{p\_av}$ values for the independent validation set. However, the prediction models constructed with spectral resolutions greater than 100 nm showed notably worse prediction performance (see Figure 8a–c).

**Table 3.** Comparison of SOM prediction accuracy values of each model with different spectral configurations.

| Configurations | Models | $RMSE_{cv\_av}$ | $R^2_{cv\_av}$ | $RPIQ_{cv\_av}$ | $RMSE_{p\_av}$ | $R^2_{p\_av}$ | $RPIQ_{p\_av}$ |
|---|---|---|---|---|---|---|---|
| ASD_1/1 | RR | 3.84 | 0.74 | 3.52 | 3.81 | 0.78 | 3.85 |
| | PLSR | 3.85 | 0.74 | 3.51 | 3.83 | 0.78 | 3.82 |
| | SVMR | 3.92 | 0.73 | 3.46 | 3.87 | 0.78 | 3.80 |
| | BPNN | 3.91 | 0.73 | 3.45 | 3.81 | 0.78 | 3.85 |
| Con_3/10 | RR | 3.84 | 0.74 | 3.51 | 3.81 | 0.78 | 3.85 |
| | PLSR | 3.85 | 0.74 | 3.51 | 3.82 | 0.78 | 3.84 |
| | SVMR | 3.93 | 0.73 | 3.45 | 3.88 | 0.77 | 3.79 |
| | BPNN | 3.89 | 0.74 | 3.47 | 3.80 | 0.78 | 3.85 |
| Con_10/10 | RR | 3.83 | 0.74 | 3.52 | 3.81 | 0.78 | 3.85 |
| | PLSR | 3.83 | 0.74 | 3.52 | 3.82 | 0.78 | 3.83 |
| | SVMR | 3.94 | 0.73 | 3.44 | 3.89 | 0.77 | 3.78 |
| | BPNN | 3.89 | 0.74 | 3.48 | 3.83 | 0.78 | 3.83 |
| Con_20/20 | RR | 3.83 | 0.74 | 3.53 | 3.81 | 0.78 | 3.85 |
| | PLSR | 3.82 | 0.74 | 3.53 | 3.80 | 0.78 | 3.85 |
| | SVMR | 3.93 | 0.73 | 3.44 | 3.87 | 0.78 | 3.80 |
| | BPNN | 3.88 | 0.74 | 3.48 | 3.83 | 0.78 | 3.83 |
| Con_40/40 | RR | 3.83 | 0.74 | 3.53 | 3.82 | 0.78 | 3.83 |
| | PLSR | 3.82 | 0.74 | 3.54 | 3.83 | 0.78 | 3.82 |
| | SVMR | 3.95 | 0.73 | 3.42 | 3.85 | 0.78 | 3.81 |
| | BPNN | 3.88 | 0.74 | 3.48 | 3.83 | 0.78 | 3.82 |
| Con_60/60 | RR | 3.82 | 0.74 | 3.54 | 3.82 | 0.78 | 3.84 |
| | PLSR | 3.82 | 0.74 | 3.54 | 3.82 | 0.78 | 3.83 |
| | SVMR | 3.94 | 0.73 | 3.43 | 3.86 | 0.78 | 3.80 |
| | BPNN | 3.87 | 0.74 | 3.49 | 3.81 | 0.78 | 3.84 |
| Con_80/80 | RR | 3.83 | 0.74 | 3.53 | 3.83 | 0.78 | 3.82 |
| | PLSR | 3.82 | 0.74 | 3.53 | 3.86 | 0.78 | 3.79 |
| | SVMR | 3.94 | 0.73 | 3.43 | 3.90 | 0.77 | 3.76 |
| | BPNN | 3.88 | 0.74 | 3.48 | 3.82 | 0.78 | 3.83 |
| Con_100/100 | RR | 3.85 | 0.74 | 3.50 | 3.83 | 0.78 | 3.82 |
| | PLSR | 3.85 | 0.74 | 3.51 | 3.86 | 0.78 | 3.79 |
| | SVMR | 3.97 | 0.73 | 3.41 | 3.94 | 0.77 | 3.73 |
| | BPNN | 3.90 | 0.73 | 3.47 | 3.83 | 0.78 | 3.82 |
| Con_150/150 | RR | 3.94 | 0.73 | 3.43 | 3.98 | 0.76 | 3.69 |
| | PLSR | 3.96 | 0.73 | 3.41 | 3.97 | 0.76 | 3.68 |
| | SVMR | 4.04 | 0.71 | 3.35 | 4.04 | 0.76 | 3.63 |
| | BPNN | 3.98 | 0.72 | 3.40 | 3.99 | 0.76 | 3.68 |
| Con_200/200 | RR | 4.09 | 0.71 | 3.31 | 4.11 | 0.75 | 3.58 |
| | PLSR | 4.09 | 0.71 | 3.31 | 4.12 | 0.75 | 3.57 |
| | SVMR | 4.09 | 0.71 | 3.31 | 4.10 | 0.75 | 3.58 |
| | BPNN | 4.09 | 0.71 | 3.30 | 4.11 | 0.75 | 3.58 |

In the second experiment, we analyzed the effect of the SNR on the SOM estimation using the RR, PLSR, SVMR, and BPNN prediction models. We assumed that the initial spectral data, as measured by the ASD spectrometer, had a SNR of 100%. To reduce the SNR of the data set from 100% to 1%, Gaussian random noise was introduced into the

spectral data. The observed influence of the spectrometer's SNR on the accuracy of the SOM estimation aligned with our expectations. The results from these four regression methods on the validation sets consistently indicated that there was no significant alteration in the RMSE, $R^2$, or RPIQ as the SNR decreased. However, as the SNR continued to drop below a threshold, a notable decrease in the prediction accuracy became apparent (Table 4). This trend was more vividly displayed through the $RMSE_{p\_av}$, $R^2_{p\_av}$, and $RPIQ_{p\_av}$ for the independent validation set (Figure 8d–f). As illustrated in Figure 8d, this threshold was approximately 15%. When the SNR exceeded 20%, any minor differences observed between groups with varying SNRs could be attributed to the slightly different hyperparameters selected for cross-validation, and this effect was relatively weak. However, when the SNR fell below 15%, the prediction performance became highly sensitive to changes in the SNR. Based on these findings, we can conclude that at lower SNR levels, the estimation performance significantly improves with increasing SNR levels. In contrast, the influence of the SNR on the SOM estimation diminishes as the SNR reaches higher levels.

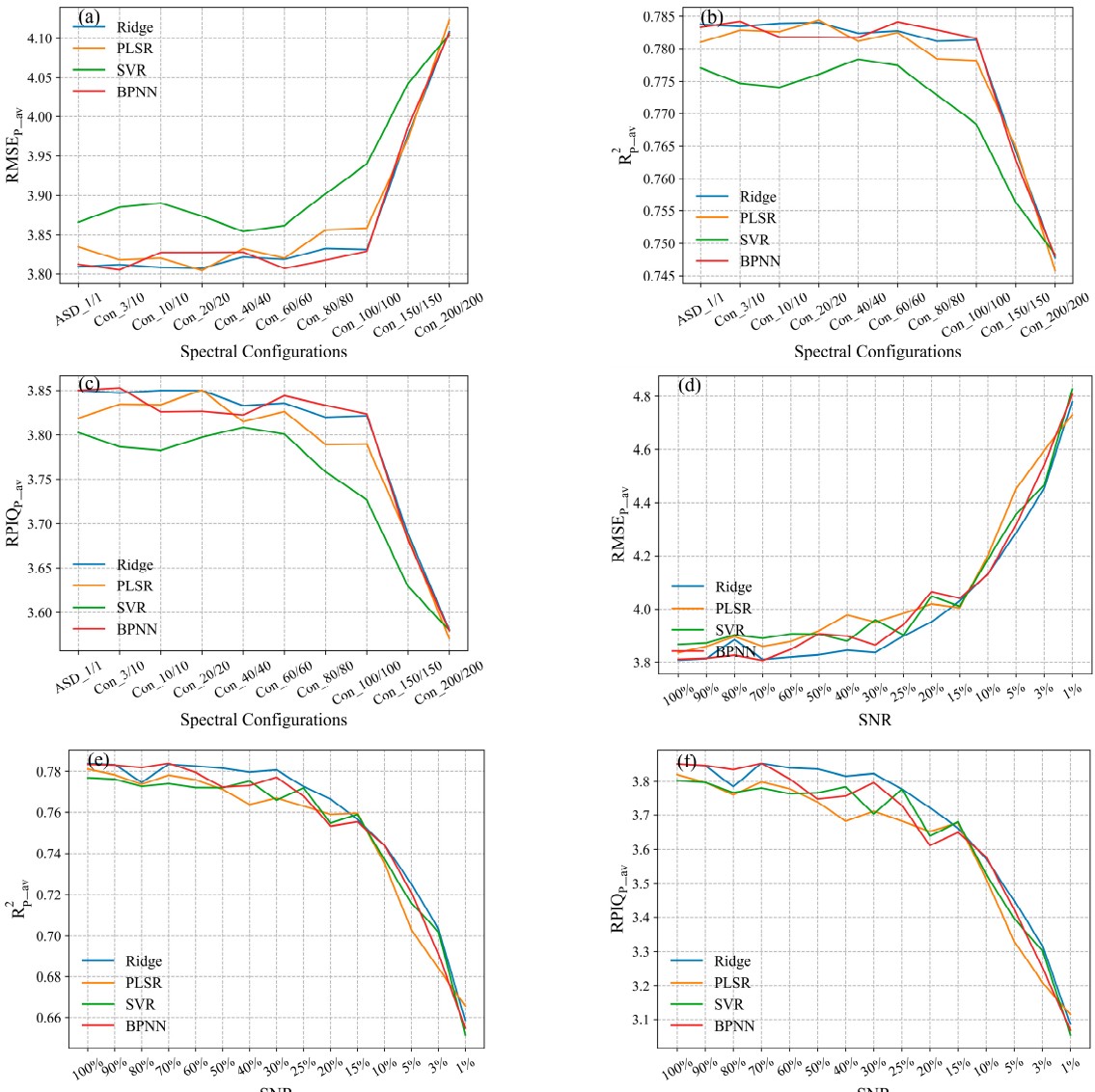

**Figure 8.** Variation in $RMSE_{p\_av}$, $R^2_{p\_av}$, and $RPIQ_{p\_av}$ with all spectral configurations and SNR levels for the four prediction models. (**a**) $RMSE_{p\_av}$ for predicting SOM in various spectral configurations; (**b**) $R^2_{p\_av}$ for predicting SOM in various spectral configurations; (**c**) $RPIQ_{p\_av}$ for predicting SOM in various spectral configurations; (**d**) $RMSE_{p\_av}$ for predicting SOM in various SNR levels; (**e**) $R^2_{p\_av}$ for predicting SOM in various SNR levels; (**f**) $RPIQ_{p\_av}$ for predicting SOM in various SNR levels.

**Table 4.** Comparison of SOM prediction accuracy values of each model with different SNR levels.

| SNR Levels | Models | $RMSE_{cv\_av}$ | $R^2_{cv\_av}$ | $RPIQ_{cv\_av}$ | $RMSE_{p\_av}$ | $R^2_{p\_av}$ | $RPIQ_{p\_av}$ |
|---|---|---|---|---|---|---|---|
| 100% SNR | RR | 3.84 | 0.74 | 3.52 | 3.81 | 0.78 | 3.85 |
| | PLSR | 3.85 | 0.74 | 3.51 | 3.83 | 0.78 | 3.82 |
| | SVMR | 3.92 | 0.73 | 3.46 | 3.87 | 0.78 | 3.80 |
| | BPNN | 3.91 | 0.73 | 3.45 | 3.81 | 0.78 | 3.85 |
| 90% SNR | RR | 3.85 | 0.74 | 3.51 | 3.81 | 0.78 | 3.84 |
| | PLSR | 3.86 | 0.74 | 3.50 | 3.86 | 0.78 | 3.80 |
| | SVMR | 3.92 | 0.73 | 3.45 | 3.87 | 0.78 | 3.80 |
| | BPNN | 3.91 | 0.73 | 3.46 | 3.82 | 0.78 | 3.85 |
| 80% SNR | RR | 3.84 | 0.74 | 3.52 | 3.88 | 0.77 | 3.79 |
| | PLSR | 3.87 | 0.74 | 3.49 | 3.90 | 0.77 | 3.76 |
| | SVMR | 3.92 | 0.73 | 3.45 | 3.90 | 0.77 | 3.77 |
| | BPNN | 3.91 | 0.73 | 3.46 | 3.83 | 0.78 | 3.83 |
| 70% SNR | RR | 3.84 | 0.74 | 3.52 | 3.81 | 0.78 | 3.85 |
| | PLSR | 3.87 | 0.74 | 3.49 | 3.86 | 0.78 | 3.80 |
| | SVMR | 3.91 | 0.73 | 3.46 | 3.89 | 0.77 | 3.78 |
| | BPNN | 3.89 | 0.73 | 3.47 | 3.81 | 0.78 | 3.85 |
| 60% SNR | RR | 3.83 | 0.74 | 3.53 | 3.82 | 0.78 | 3.84 |
| | PLSR | 3.87 | 0.74 | 3.49 | 3.88 | 0.78 | 3.78 |
| | SVMR | 3.91 | 0.73 | 3.46 | 3.91 | 0.77 | 3.76 |
| | BPNN | 3.87 | 0.74 | 3.49 | 3.85 | 0.78 | 3.81 |
| 50% SNR | RR | 3.86 | 0.74 | 3.50 | 3.83 | 0.78 | 3.83 |
| | PLSR | 3.90 | 0.73 | 3.46 | 3.92 | 0.77 | 3.74 |
| | SVMR | 3.91 | 0.73 | 3.46 | 3.91 | 0.77 | 3.77 |
| | BPNN | 3.92 | 0.73 | 3.45 | 3.91 | 0.77 | 3.75 |
| 40% SNR | RR | 3.87 | 0.74 | 3.49 | 3.84 | 0.78 | 3.81 |
| | PLSR | 3.90 | 0.73 | 3.46 | 3.98 | 0.76 | 3.68 |
| | SVMR | 3.92 | 0.73 | 3.45 | 3.88 | 0.78 | 3.78 |
| | BPNN | 3.93 | 0.73 | 3.44 | 3.90 | 0.77 | 3.76 |
| 30% SNR | RR | 3.85 | 0.74 | 3.50 | 3.84 | 0.78 | 3.82 |
| | PLSR | 3.88 | 0.74 | 3.48 | 3.95 | 0.77 | 3.71 |
| | SVMR | 3.92 | 0.73 | 3.45 | 3.96 | 0.77 | 3.70 |
| | BPNN | 3.88 | 0.74 | 3.48 | 3.86 | 0.78 | 3.80 |
| 25% SNR | RR | 3.97 | 0.72 | 3.41 | 3.90 | 0.77 | 3.78 |
| | PLSR | 3.95 | 0.73 | 3.42 | 3.98 | 0.76 | 3.68 |
| | SVMR | 3.95 | 0.73 | 3.42 | 3.90 | 0.77 | 3.78 |
| | BPNN | 4.00 | 0.72 | 3.38 | 3.94 | 0.77 | 3.73 |
| 20% SNR | RR | 3.97 | 0.72 | 3.40 | 3.95 | 0.77 | 3.72 |
| | PLSR | 3.96 | 0.72 | 3.41 | 4.02 | 0.76 | 3.65 |
| | SVMR | 4.00 | 0.72 | 3.38 | 4.05 | 0.75 | 3.64 |
| | BPNN | 4.04 | 0.71 | 3.35 | 4.06 | 0.75 | 3.61 |
| 15% SNR | RR | 4.07 | 0.71 | 3.32 | 4.03 | 0.76 | 3.66 |
| | PLSR | 4.06 | 0.71 | 3.33 | 4.01 | 0.76 | 3.68 |
| | SVMR | 4.05 | 0.71 | 3.34 | 4.01 | 0.76 | 3.68 |
| | BPNN | 4.11 | 0.70 | 3.29 | 4.04 | 0.76 | 3.65 |
| 10% SNR | RR | 4.17 | 0.70 | 3.25 | 4.13 | 0.74 | 3.57 |
| | PLSR | 4.17 | 0.70 | 3.25 | 4.20 | 0.74 | 3.51 |
| | SVMR | 4.17 | 0.69 | 3.25 | 4.19 | 0.74 | 3.53 |
| | BPNN | 4.22 | 0.69 | 3.21 | 4.13 | 0.74 | 3.58 |
| 5% SNR | RR | 4.36 | 0.67 | 3.10 | 4.28 | 0.73 | 3.45 |
| | PLSR | 4.33 | 0.68 | 3.12 | 4.45 | 0.70 | 3.33 |
| | SVMR | 4.38 | 0.67 | 3.09 | 4.36 | 0.72 | 3.40 |
| | BPNN | 4.43 | 0.66 | 3.06 | 4.32 | 0.72 | 3.42 |
| 3% SNR | RR | 4.55 | 0.65 | 2.97 | 4.45 | 0.70 | 3.32 |
| | PLSR | 4.51 | 0.65 | 3.00 | 4.60 | 0.68 | 3.21 |
| | SVMR | 4.53 | 0.65 | 2.99 | 4.46 | 0.70 | 3.30 |
| | BPNN | 4.61 | 0.64 | 2.93 | 4.54 | 0.69 | 3.25 |
| 1% SNR | RR | 4.83 | 0.60 | 2.80 | 4.78 | 0.66 | 3.09 |
| | PLSR | 4.76 | 0.61 | 2.84 | 4.73 | 0.67 | 3.12 |
| | SVMR | 4.85 | 0.60 | 2.79 | 4.83 | 0.65 | 3.06 |
| | BPNN | 4.84 | 0.60 | 2.79 | 4.81 | 0.65 | 3.07 |

In our current study, the effect of lower SNR levels on the performance of the SOM estimation was more pronounced compared to the spectral resolutions. The results from the independent validation set reveal that the models constructed using the spectral configuration Con_200/200 yielded subpar prediction performance, with an $RMSE_{p\_av} > 4.10$, an $R^2_{p\_av} < 0.75$, and an $RPIQ_{p\_av} < 3.57$ (Table 3). The prediction models using a 1%

SNR level exhibited the poorest prediction performance, with an $RMSE_{p\_av} > 4.83$, an $R^2_{p\_av} < 0.65$, and an $RPIQ_{p\_av} < 3.06$ (Table 4). Furthermore, all four models (RR, PLSR, SVMR, and BPNN) built using the spectral configuration library and the spectral SNR library displayed similar trends. However, it remains unclear whether these four models yielded significantly different results for the SOM estimation. While Figure 8 provides an initial insight into the impact of the spectral resolution and SNR on the SOM estimation, it does not ascertain whether this impact is statistically significant. Therefore, ANOVA was employed to conduct a more in-depth analysis to identify the factors contributing to the effect on the SOM estimation accuracy.

*3.6. ANOVA Results*

In our current study, we used ANOVA techniques to identify significant factors influencing the SOM estimation and to determine the levels of these factors that contributed to differences. Since $RPIQ = IQ/RMSE$, where IQ represents the difference between the third and first quartiles of the true SOM values for each group, it introduces errors into the dataset, thus leading to larger within-group errors and affecting the analysis results. Therefore, we focused on conducting ANOVA with the two performance indicators, $RMSE_p$ and $R^2_p$. In our study, we employed four regression models to predict the SOM for each spectral configuration and SNR level. Initially, we applied two-way ANOVAs to analyze the contributions of the models and instrument parameters to the estimation of the SOM. Subsequently, one-way ANOVAs were performed under each of the four prediction models to investigate whether the spectral resolutions and SNRs had significant effects on the SOM estimation. It is essential to note that for ANOVA to be valid, the data must meet homogeneity and normal distribution requirements. Our analysis indicates that the variances of the residuals for the $RMSE_p$ and $R^2_p$ in each group were equal, as was demonstrated by Levene's test. The significance of the Shapiro–Wilks test was greater than 0.05, thereby indicating that the data satisfied normality requirements and supported the use of ANOVA.

3.6.1. Two-Way ANOVA Results

We considered the two performance indicators, $RMSE_p$ and $R^2_p$, as the dependent variables and examined the factors of the spectral configurations and prediction models. Table 5 presents the results of the two-way ANOVA for the spectral configurations and models. When the significance level was set at 0.05, it became evident that both the spectral configurations and prediction models had a significant effect on prediction performance ($p < 0.05$; $F > F_{crit}$). However, their interaction did not appear to be significant ($p > 0.05$; $F > F_{crit}$). Notably, the $p$ value associated with the spectral configurations was considerably greater than the $p$ value linked to the prediction models. This suggests that the spectral configuration has a more substantial impact on the prediction performance and provides stronger grounds for rejecting the null hypothesis. These findings, in conjunction with the conclusions drawn in Section 3.3, underscore that the spectral resolution has a greater effect on the prediction performance of the SOM compared to the prediction model used.

Table 6 presents the results of a two-way ANOVA in which the $RMSE_p$ and $R^2_p$ were considered as the dependent variables. The SNR levels and prediction models were treated as the two factors. With a significance level set at 0.05, it became evident that the main effects of the SNR levels and prediction models were statistically significant, while their interaction effects were not. The $p$ value associated with the SNR levels was significantly larger than the $p$ value linked to the prediction models, thus indicating that the spectrometer's SNR had a more substantial impact on the SOM prediction performance.

**Table 5.** Examining the outcomes of a two-way ANOVA involving $RMSE_p$ and $R^2_p$ as dependent variables, with spectral configurations and prediction models considered as the influencing factors.

| Indicator | Source | SS | df | MS | F | *p* Value | $F_{crit}$ |
|---|---|---|---|---|---|---|---|
| | Configuration | 6.50 | 9 | 0.72 | 18.96 | $9.51 \times 10^{-29}$ | 1.89 |
| | Model | 0.49 | 3 | 0.16 | 4.33 | 0.0049 | 2.62 |
| $RMSE_p$ | Interaction | 0.16 | 27 | 0.0058 | 0.15 | 1.00 | 1.50 |
| | Error | 28.96 | 760 | 0.038 | | | |
| | Total | 36.11 | 799 | | | | |
| | Configuration | 0.094 | 9 | 0.010 | 21.10 | $5.48 \times 10^{-32}$ | 1.89 |
| | Model | 0.0072 | 3 | 0.0024 | 4.86 | 0.0023 | 2.62 |
| $R^2_p$ | Interaction | 0.0022 | 27 | $8.32 \times 10^{-5}$ | 0.17 | 1.00 | 1.50 |
| | Error | 0.38 | 760 | | | | |
| | Total | 0.48 | 799 | | | | |

SS: Sum of squares; df: degree of freedom; MS: mean square.

**Table 6.** Examining the outcomes of a two-way ANOVA involving $RMSE_p$ and $R^2_p$ as dependent variables, with SNR levels and prediction models considered as the influencing factors.

| Indicator | Source | SS | df | MS | F | *p* Value | $F_{crit}$ |
|---|---|---|---|---|---|---|---|
| | SNR | 91.82 | 14 | 6.56 | 79.05 | $8 \times 10^{-157}$ | 1.70 |
| | Model | 0.75 | 3 | 0.25 | 3.02 | 0.029 | 2.61 |
| $RMSE_p$ | Interaction | 1.14 | 42 | 0.027 | 0.33 | 1.00 | 1.39 |
| | Error | 94.57 | 1140 | 0.083 | | | |
| | Total | 188.29 | 1199 | | | | |
| | SNR | 1.51 | 14 | 0.11 | 92.63 | $9.9 \times 10^{-177}$ | 1.70 |
| | Model | 0.011 | 3 | 0.0036 | 3.10 | 0.026 | 2.61 |
| $R^2_p$ | Interaction | 0.019 | 42 | 0.0004 | 0.38 | 1.00 | 1.39 |
| | Error | 1.33 | 1140 | 0.0012 | | | |
| | Total | 2.86 | 1199 | | | | |

The results obtained from the two-way ANOVA lead us to the conclusion that both the instrument paraments and prediction models indeed exert a significant influence on the accuracy of SOM estimation. Furthermore, it is evident that the instrument parameters had a more pronounced impact, thereby underscoring the practical significance of our study. It is worth noting that the primary objective of this paper was to assess the variation in the SOM estimation accuracy under different spectral resolutions and SNR levels. Therefore, we performed one-way ANOVA under the various prediction models to further scrutinize the effect of the instrument parameters on the accuracy of the SOM estimation.

### 3.6.2. One-Way ANOVA Results

The present analysis includes one-way ANOVA results for the spectral configurations and SNR levels, as displayed in Tables 7 and 8, respectively. ANOVA was conducted at a 95% confidence level, with a significance level set at 0.05. For all multiple regression models, the outcomes in Tables 7 and 8 indicate that both the ANOVA for the spectral configuration and the ANOVA for the SNR level yielded *p* values below 0.05. According to ANOVA principles, if the *p* value is less than 0.05, and the calculated F-value exceeds the critical F value ($F_{crit}$), the null hypothesis is rejected. Therefore, our findings suggest that, regardless of the choice of prediction models, the spectrometer's parameters significantly influenced the accuracy of the SOM prediction. Furthermore, as ANOVA alone does not specify which groups differ from one another, post hoc tests using the Tukey method were employed to further interpret the results of the ANOVA, which specifically focused on the $R^2_p$. In this study, post hoc tests were conducted using Python 3.8.5 and the statsmodels package.

**Table 7.** Results of one-way ANOVA using spectral configurations as the independent variables.

| Model | Indicator | Source | SS | df | MS | F | *p* Value | $F_{crit}$ |
|---|---|---|---|---|---|---|---|---|
| RR | $RMSE_p$ | Configuration | 1.81 | 9 | 0.20 | 5.96 | $2.42 \times 10^{-7}$ | 1.93 |
| | | Error | 6.42 | 190 | 0.034 | | | |
| | | Total | 8.23 | 199 | | | | |
| | $R^2_p$ | Con | 0.026 | 9 | 0.002 | 6.84 | $1.69 \times 10^{-8}$ | 1.93 |
| | | Error | 0.081 | 190 | 0.0004 | | | |
| | | Total | 0.11 | 199 | | | | |
| PLSR | $RMSE_p$ | Con | 1.79 | 9 | 0.20 | 7.46 | $2.61 \times 10^{-9}$ | 1.93 |
| | | Error | 5.06 | 190 | 0.027 | | | |
| | | Total | 6.85 | 199 | | | | |
| | $R^2_p$ | Con | 0.026 | 9 | 0.0029 | 7.57 | $1.84 \times 10^{-9}$ | 1.93 |
| | | Error | 0.073 | 190 | 0.0004 | | | |
| | | Total | 0.099 | 199 | | | | |
| SVMR | $RMSE_p$ | Con | 1.29 | 9 | 0.14 | 2.39 | 0.014 | 1.93 |
| | | Error | 11.42 | 190 | 0.060 | | | |
| | | Total | 12.72 | 199 | | | | |
| | $R^2_p$ | Con | 0.018 | 9 | 0.002 | 2.69 | 0.0058 | 1.93 |
| | | Error | 0.14 | 190 | 0.0007 | | | |
| | | Total | 0.16 | 199 | | | | |
| BPNN | $RMSE_p$ | Con | 1.82 | 9 | 0.22 | 6.48 | $4.97 \times 10^{-8}$ | 1.93 |
| | | Error | 5.93 | 190 | 0.031 | | | |
| | | Total | 7.75 | 199 | | | | |
| | $R^2_p$ | Con | 0.026 | 9 | 0.0029 | 7.25 | $4.78 \times 10^{-9}$ | 1.93 |
| | | Error | 0.076 | 190 | 0.0004 | | | |
| | | Total | 0.10 | 199 | | | | |

**Table 8.** Results of one-way ANOVA using SNR level as the independent variable.

| Model | Indicator | Source | SS | df | MS | F | *p* Value | $F_{crit}$ |
|---|---|---|---|---|---|---|---|---|
| RR | $RMSE_p$ | SNR | 23.15 | 14 | 1.65 | 20.07 | $7.2 \times 10^{-35}$ | 1.73 |
| | | Error | 23.47 | 285 | 0.082 | | | |
| | | Total | 46.62 | 299 | | | | |
| | $R^2_p$ | SNR | 0.38 | 14 | 0.027 | 23.88 | $3.21 \times 10^{-40}$ | 1.73 |
| | | Error | 0.32 | 285 | 0.0011 | | | |
| | | Total | 0.69 | 299 | | | | |
| PLSR | $RMSE_p$ | SNR | 22.95 | 14 | 1.64 | 21.60 | $4.49 \times 10^{-37}$ | 1.73 |
| | | Error | 21.62 | 285 | 0.076 | | | |
| | | Total | 44.57 | 299 | | | | |
| | $R^2_p$ | SNR | 0.38 | 14 | 0.027 | 24.73 | $2.31 \times 10^{-41}$ | 1.73 |
| | | Error | 0.31 | 285 | 0.0011 | | | |
| | | Total | 0.69 | 299 | | | | |
| SVMR | $RMSE_p$ | SNR | 22.02 | 14 | 1.57 | 16.49 | $2.17 \times 10^{-29}$ | 1.73 |
| | | Error | 27.19 | 285 | 0.095 | | | |
| | | Total | 49.21 | 299 | | | | |
| | $R^2_p$ | SNR | 0.36 | 14 | 0.026 | 19.80 | $1.8 \times 10^{-34}$ | 1.73 |
| | | Error | 0.37 | 285 | 0.0013 | | | |
| | | Total | 0.74 | 299 | | | | |
| BPNN | $RMSE_p$ | SNR | 24.85 | 14 | 1.77 | 22.69 | $1.35 \times 10^{-38}$ | 1.73 |
| | | Error | 22.29 | 285 | 0.078 | | | |
| | | Total | 47.14 | 299 | | | | |
| | $R^2_p$ | SNR | 0.41 | 14 | 0.029 | 26.03 | $4.69 \times 10^{-43}$ | 1.73 |
| | | Error | 0.31 | 285 | 0.0011 | | | |
| | | Total | 0.72 | 299 | | | | |

Figure 9 visualizes the results of the post hoc test of the spectral configurations, with 1 indicating a significant difference between the two groups and 0 signifying no significant difference. While the results of the Tukey test exhibited slight discrepancies across the different prediction models, a consistent observation emerged: there was no notable distinction between the $R^2_p$ values for a 100 nm spectral resolution and those for higher spectral resolutions. The variations observed among the groups are primarily attributed to random factors introduced by the data, thereby mirroring the conclusions drawn from the visual assessments in Figure 8a–c. Consequently, based on the aforementioned post hoc test results, we can confidently assert that a similar SOM estimation performance can be attained when employing a spectral resolution within 100 nm.

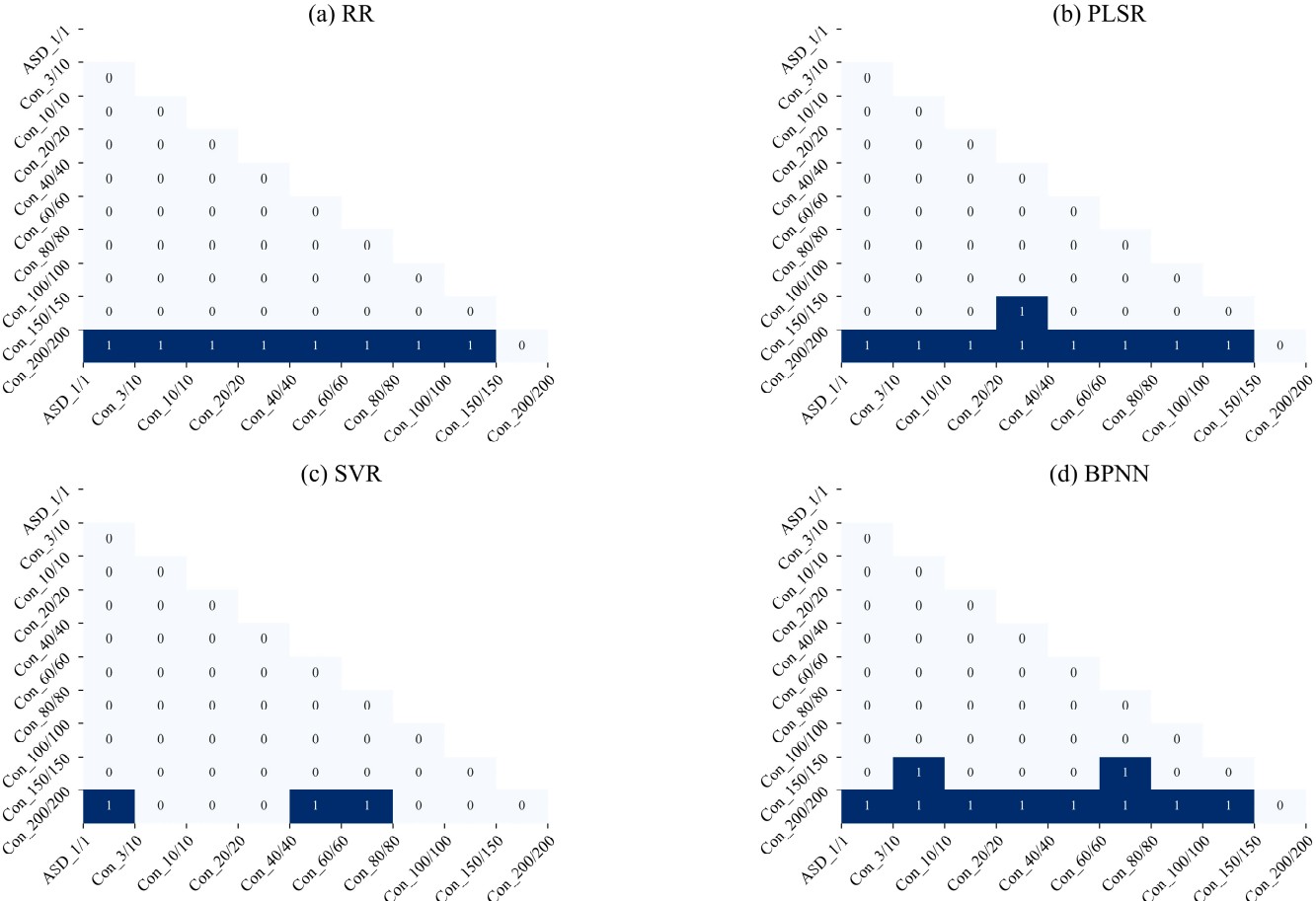

**Figure 9.** Results of post hoc test for each set of $R^2_p$ values regarding spectral configurations (1 indicates significant difference between groups; 0 indicates no significant difference).

Figure 10 shows the post hoc test results for $R^2_p$ across the four prediction models at various SNR levels. Across the 15 distinct sets of SNR levels, all four models exhibited a significant difference in prediction performance at the 10% threshold. Notably, there was no significant difference in the $R^2_p$ values from 100% to 15%, thus aligning with the pattern observed through visual inspection in Figure 8d–f. This pattern underscores that excessive noise in the spectral data can obscure the characteristic information of the measured object, thereby ultimately leading to reduced estimation accuracy.

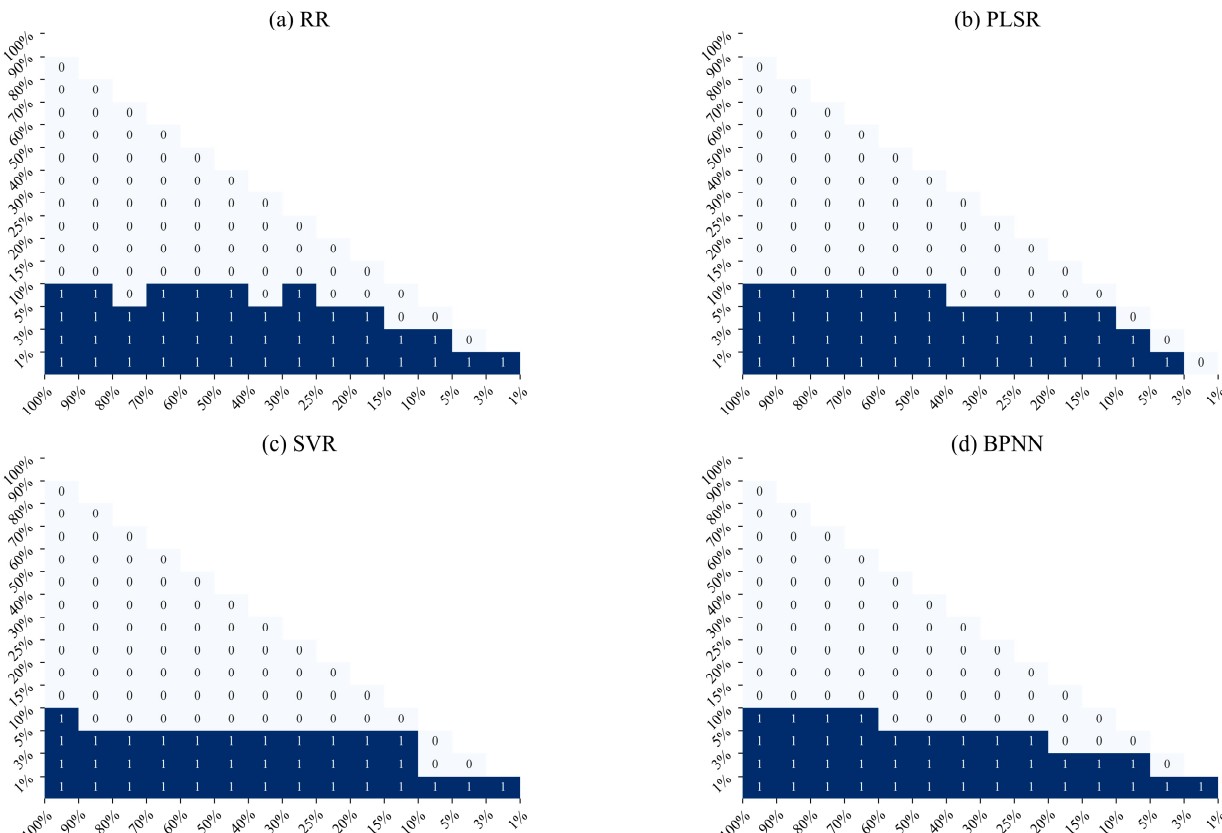

**Figure 10.** Results of post hoc test for each set of $R^2_p$ values regarding SNR levels (1 indicates significant difference between groups; 0 indicates no significant difference.

## 4. Discussion

The spectral resolution and SNR represent two crucial parameters in optical remote sensing payloads. In our study, we ascertained that modeling approaches, spectral resolution, and SNRs collectively influence the SOM estimation through an analysis and comparison of the results derived from the same original soil spectral library. The extent of these factors' impacts on the estimation performance has been previously elucidated. In this section, we delve into how instrument parameters influence estimation performance and provide pertinent recommendations for designing optical instruments dedicated to SOM monitoring.

The SNR of spectrometers is intricately linked to the optical subsystem, detector subsystem, and electronic subsystem. In practical spectrometer operation, the maximum achievable SNR in a specific spectral channel is constrained by fluctuations in the detector dark current, the noise associated with readout processes, the photon noise originating from thermal emissions within the instrument, the nonuniformities within the detector array, and the challenges related to detector calibration [39]. In the field, reduced solar irradiance within the NIR region similarly diminishes the SNR within the NIR spectral bands. Consequently, the SNR of spectroscopic instruments is affected by numerous practical factors. In the actual design of a spectrometer, there exist trade-offs among these parameters. Generally, enhancing the SNR leads to compromises such as reduced spectral resolution and increased dimensions and weight of the optical components. [17,25].

The spectral configuration Con_200/200 provided poor prediction performance, as indicated by an $RMSE_{p\_av} > 4.10$, an $R^2_{p\_av} < 0.75$, and an $RPIQ_{p\_av} < 3.57$. The prediction models using the 1% SNR level produced the worst prediction performance, with an $RMSE_{p\_av} > 4.83$, an $R^2_{p\_av} < 0.65$, and an $RPIQ_{p\_av} < 3.06$. When taken in conjunction with the results of the ANOVA, it becomes evident that the SNR tends to exert a more pronounced influence on SOM estimation compared to other factors. The phenomenon

observed is that the prediction accuracy increases with higher SNR levels, but the rate of improvement diminishes at elevated SNR levels. It is well known that the fundamental principle underlying most machine learning methods involves regressors learning data features and iteratively minimizing losses to make predictions on new data samples. Excessive noise can overpower the inherent data features, thereby resulting in reduced prediction accuracy. This is the reason why the SNR significantly impacts the prediction performance. Furthermore, hyperspectral data typically encompass a multitude of wavelength variables and, in practice, are susceptible to the challenges associated with the "curse of dimensionality" [40]. Our research suggests that decreasing the spectral resolution by reducing the number of spectral bands could alleviate the issues related to multicollinearity and redundant information. This simplification of the prediction models can yield comparable or even more efficient results compared to higher spectral resolutions. Nevertheless, an excessively low number of spectral bands would result in insufficient valid information for accurately estimating the SOM.

When considering optical remote sensing instruments intended for SOM monitoring, it is advisable to prioritize addressing the design requirement for the SNR to mitigate its influence on the precise estimation of the SOM. In certain situations, opting for a lower spectral resolution may be suitable to meet the spectrometer's SNR prerequisites. Additionally, for hyperspectral instruments, reducing the spectral resolution also entails a decrease in the number of bands, which equates to a reduction in the volume of the spectral data. This approach also simplifies data processing, thus making it less cumbersome. Consequently, investigating the impact of the spectral resolution and SNR levels on the SOM estimation enhances the potential for utilizing spectroscopic techniques for SOM estimation. This applicability can extend to various soil regions, and our findings can be extrapolated to broader scales.

## 5. Conclusions

In this study, we simulated two spectral libraries using the spectra measured by an ASD spectrometer. We evaluated the effect of the two fundamental spectrometer parameters—the spectral resolution and SNR—on the estimation of the SOM using four multivariate regression methods (i.e., RR, PLSR, SVMR, and BPNN). The following conclusions are derived from our experimental analysis:

1.  Various spectral resolutions, varying SNR levels, and the utilization of distinct multivariate regression models for prediction all exert noteworthy influences on the SOM prediction performance. Notably, the variations in the prediction performance attributed to instrument parameters surpassed those attributed to the prediction models. The most substantial disparities stem from SNR levels, with the spectral resolution differences following closely behind.
2.  The ANOVA analysis of the performance indicator $R^2_p$ indicates that there is no significant discrepancy in the SOM prediction performance when the spectral resolution falls within 100 nm. However, when the spectral resolution exceeds 100 nm, a significant decline in the estimation accuracy becomes evident.
3.  The SNR level of the spectroscopic instruments emerged as the most pivotal factor for the precise estimation of the SOM. Typically, higher SNR levels correspond to enhanced estimation accuracy. Nevertheless, as the SNR reaches higher levels, its impact on the SOM estimation diminishes. The ANOVA results for $R^2_p$ suggest that when the SNR level surpasses 15%, it no longer yields a significant difference in the SOM estimation performance.

In conclusion, the spectral resolution and SNR are important indicators of a spectrometer. Grasping the influence of these parameters on the SOM estimation paves the way for the efficient design of optical remote sensing payloads aimed at monitoring large-scale SOM variations in the future.

**Author Contributions:** Conceptualization, B.Y.; methodology, B.Y.; software, B.Y.; validation, B.Y., J.Y. and C.Y.; formal analysis, B.Y.; investigation, B.Y.; resources, B.Y., J.X., C.M. and H.D.; data curation, B.Y.; writing—original draft preparation, B.Y.; writing—review and editing, C.Y., J.Y., J.X. and C.M.; visualization, B.Y. and J.Y.; supervision, C.Y. and J.Y.; project administration, C.Y.; funding acquisition, C.Y. All authors have read and agreed to the published version of the manuscript.

**Funding:** This research was funded by the Jilin Key R&D Program of China under Grant 20230201036GX; it was also funded in part by the National Natural Science Foundation of China (NSFC) under Grant 62105331, and Grant 62275114; in part by the Qingdao Industrial Experts Program; and in part by the Taishan Industrial Experts Program.

**Data Availability Statement:** The data is not available, because the team data involves privacy issues.

**Acknowledgments:** The editor and the reviewers are thanked for their helpful comments and criticisms of the initial draft of this paper.

**Conflicts of Interest:** The authors declare no conflict of interest.

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
