# Peer review of "Impact of Spectral Resolution and Signal-to-Noise Ratio in Vis–NIR Spectrometry on Soil Organic Matter Estimation"

_remotesensing, doi:10.3390/rs15184623_

Round 1
Reviewer 1 Report
In this study, authors used field measurments to discuss the effect of spectral resolution and SNR on SOM estimation. The results showed that SNR has a significant effects on SOM estimation when SNR less than 15%. And this research provide a refere for future optical remote sensing payloads for SOM monitoring. But there are some problems and need to revise.
1. A boundary of the Heilongjiang Province was wrong in eagle-eye map, Please check and revised it.
2. The results of the modeling was missed, such as statistical analysis, variable selection results of the statistical modeling, and so on.
3.Parameter setting of neural network model were missed, too.
3. Results and discussion should be separated.
4. Some results and figures can be moved to attachment, such as figure 6 and figure 7. and related results.
Other comments,
1. The abbr. of SNR, was appeared first time in the abstract, should give the full name. The same problem SNR (Line 71).
2. The title name of the column was not matched with the contect in table 1(SSI SR...).
3. the title of y-axes was missed in figure 4.
4. Delete the text of (a) in figure 5.
5. the information from 450 to 775 was missed in figure 6 of Con_100/100, Con_150/150, Con_200/200.
Reviewer 2 Report
The effects of spectral resolution and signal-to-noise ratio of spectral instruments on soil organic matter estimation are discussed. The research content is innovative and valuable. However, there are still some problems in this paper that need to be modified.
1 There are some grammatical errors in the paper, so it is suggested to optimize the writing of the paper. For example, line 16: “most studies focuses on using...”; Line 19, SNR is the first time appearing, the full name is needed.
2 In table 1, row 2, here the setting principle of 3 and 10 is not explained
3 Spectral pretreatment methods can significantly affect prediction accuracy. However, it is not stated whether the spectral data in the experiment has been preprocessed, and if it has not been preprocessed, the reason why it is not necessary should be stated. If different pretreatment methods are added, the effects of spectral resolution and signal-to-noise ratio on model construction results may be more complete and convincing.
4 Are the results of this paper applicable to larger and smaller scales/other regions, and are the conclusions generalizable?
5 The experimental design of noise addition need further improvement. Is the noise added in this article representative? Is this noise too ideal? After adding the noise in this paper, we can see that the prediction effect is also very good when the SNR is very low.
6 Figure 1 needs to be beautified, and generally only one scale is needed.

-
English writing needs improvement
Reviewer 3 Report
Dear authors,
Thanks for your comprehensive study. It is interesting to read there are thesholds of SNR (15%) and spectral resolution (100 nm) of spectroscopic instruments at field scale for accuracy of SOM estimation. Prospects are probably to apply the same approach for other widely-studied land covers such as vegetation and propose a simpler and cheaper design of hyperspectral instruments for industry and sciencie purposes- at least when it comes to SOM for instance.
I believe the study is close to publication and I provide few comments to improve the readibility, because it is not easy to follow the synthetic spectral libraries you created.

Author Response
Please see the attachment, This is in response to your comments in the old version you gave me, our latest manuscript has been submitted

Reviewer 4 Report
The authors assessed the significant factors affecting the predictive performance of SOM using a spectrum instrument and found the SNR level can be the most significant factor using ANOVA analysis for prediction accuracy. The results provided great guidance for further improvement on the SOM measurement using remote sensing techniques. There are some more specific comments on the manuscript.
1. Please include some more sentences to emphasize the importance of SOM measurement on the agricultural industry in the introduction part.
2. Line 129. Provide more information about the methods for the “chemical determination” in laboratory.
3. Can spectral data measured in the laboratory correspond to spectra measured in remotely sensed data? I saw that you used a method to simulate but does this method was widely accepted? Please provide some references to justify your procedure.
4. Line 451. Improve the sentence, you do not need to highlight the significant level you used here. It will make readers confused. You may just mention it has p < .05.
5. Leave a space between in-text citations and sentences.
1. Some sentences need to improve for better readability.
2. It is not necessary to repeatedly mention common sense in the research field
